Exosomes from adipose-derived stem cells alleviate premature ovarian failure via blockage of autophagy and AMPK/mTOR pathway

Ren Yu 1
He Jinying 2
Wang Xiao 3
Liang Hongyu 1
Ma Yuzhen mayz@imnu.edu.cn 2
1 Department of Scientific Research, Inner Mongolia People’s Hospital , Hohhot , China
2 Reproductive Medicine Centre, Inner Mongolia People’s Hospital , Hohhot , China
3 Endoscopy Center, Inner Mongolia People’s Hospital , Hohhot , China
Qin Jiangjiang
Electronic publication date: 2023 Dec 14
Publication date: 2023
Volume: 11
Electronic Location ID: e16517
Received 2023 Jun 22; Accepted 2023 Nov 3
Copyright: ©2023 Ren et al.
Copyright year: 2023
Copyright holder: Ren et al.
License: This is an open access article distributed under the terms of the Creative Commons Attribution License, which permits unrestricted use, distribution, reproduction and adaptation in any medium and for any purpose provided that it is properly attributed. For attribution, the original author(s), title, publication source (PeerJ) and either DOI or URL of the article must be cited.
License URL: https://creativecommons.org/licenses/by/4.0/

Keywords: Adipose-derived stem cell, AMPK/mTOR pathway, Autophagy, Exosome, Premature ovarian failure

Funding: Inner Mongolia Autonomous Region Natural Science Fund 2020MS08154 Inner Mongolia Autonomous Region Health Science and Technology Project 202202005 This work was supported by the Inner Mongolia Autonomous Region Natural Science Fund under Grant number 2020MS08154, and the Inner Mongolia Autonomous Region Health Science and Technology Project under Grant number 202202005. The funders had no role in study design, data collection and analysis, decision to publish, or preparation of the manuscript.

==============================
Objective

The objective of this study was to investigate the effects and mechanisms of adipose-derived stem cell-derived exosome (ADSCs-Exo) in treating premature ovarian failure (POF).

Methods

We constructed a POF mouse model through intraperitoneal injection of cyclophosphamide, followed by the administration of the autophagy inhibitor 3-methyladenine (3-MA). Pathological injury, follicle stimulating hormone (FSH), malondialdehyde (MDA), reactive oxygen species (ROS), estradiol (E2), superoxide dismutase (SOD), granulosa cell (GC) apoptosis, and autophagy were assessed. Exosomes isolated from ADSCs were used to treat POF in mice. The AMPK-mTOR pathway and its proteins (p-AMPK and p-mTOR) were evaluated. A POF cell model was established using cyclophosphamide-treated human ovarian granulosa-like tumor (KGN) cells. We administered ADSCs-Exo and rapamycin to validate the mechanism of ADSCs-Exo against POF.

Results

In POF mice, 3-MA treatment attenuated pathological injuries, decreased FSH, MDA, and ROS levels, and increased E2 and SOD levels. 3-MA treatment also inhibited GC apoptosis and autophagy. ADSCs-Exo alleviated pathological injuries, improved ovarian morphology and function, and reduced oxidative stress in POF mice. ADSCs-Exo inhibited GC apoptosis and autophagy. ADSCs-Exo downregulated the expression of AMPK/mTOR pathway proteins (p-AMPK and p-mTOR). In the POF cell model, ADSCs-Exo and rapamycin inhibited AMPK/mTOR-mediated autophagy.

Conclusion

ADSCs-Exo inhibits POF through the inhibition of autophagy and the AMPK/mTOR pathway. This study provides a potential target for the clinical treatment of POF.

Introduction

Premature ovarian failure (POF) is a clinical syndrome characterized by the premature decline of ovarian function, including abnormal menstruation, elevated serum gonadotropin levels, and decreased estrogen (Yeganeh et al., 2019). POF is well recognized as a risk factor for ovarian and other reproductive cancers (Schover, 2014). In the female reproductive system, the ovary is extremely sensitive to chemotherapeutic drugs. Long-term high-dose chemotherapy will destroy the ovarian tissue and cause ovarian dysfunction, resulting in POF. Clinically, it is mainly manifested in the early occurrence of amenorrhea, infertility, menopausal syndrome, and aging diseases, which seriously affects the quality of life and self-confidence of young women (Blumenfeld, 2019; Kawamura, Kawamura & Hsueh, 2016). At present, the clinical treatment of POF is mainly to maintain menstruation through hormone replacement therapy and alleviate the related symptoms caused by estrogen deficiency. However, due to the irreversibility of ovarian failure, to our knowledge, no effective method can restore the ovarian function of patients with POF (Liu, Li & Yang, 2019). Therefore, there is an urgent need to focus on the underlying mechanisms for the etiopathogenesis of POF.

The pathogenesis of POF is mainly related to the obstacle of follicular maturation, abnormal activation, atresia of primordial follicles, and the increase of oocyte granulosa cell (GC) apoptosis. Autophagy is a ubiquitous self-stabilizing mechanism in eukaryotic cells (Levine & Kroemer, 2019). In recent years, autophagy has attracted much attention in the regulation of ovarian function, which plays a positive or negative regulatory role in follicular development and atresia (Bhardwaj et al., 2021). Liu et al. (2020b) found that antagonism of autophagy significantly inhibits GC proliferation (Zhou et al., 2017). Sun et al. (2021) confirmed that oxidative stress induced autophagic cell death in mouse ovarian GCs. Exogenous administration of follicle stimulating hormone (FSH) plays a protective role in anovulatory disorders by inhibiting GC (Shen et al., 2017). Choi et al. (2010) showed that gonadotropin promotes the accumulation of autophagy and further induces GCs apoptosis, thereby facilitating ovarian follicular development and atresia in rats. However, the role of autophagy in chemotherapeutic drug-induced POF is unclear.

Adipose-derived stem cells (ADSCs), sourced from adipose tissue, possess the versatility to differentiate into various cell and tissue types under appropriate induction conditions (Zhou, Wei & He, 2021). ADSCs-derived exosomes (ADSCs-Exo) are important pathways and modalities for ADSCs to exert their effects (Zhang et al., 2021). Literature shows that in vitro stem cells promote the survival and development of early mouse cocultured follicles by secreting cytokines (Sheikhansari et al., 2018; Zheng et al., 2019). Placental-derived stem cells improve POF induced by chemotherapeutic drugs and promote follicular development through paracrine mechanism (Ling et al., 2019). Chinese scholars reported for the first time that exosomes derived from bone marrow mesenchymal stem cells have a protective effect on POF in rats (Liu et al., 2020a; Sun et al., 2019). Exosomes regulate the development of diseases by transferring and releasing functional molecules. GATA3 released from tumor-associated macrophage-derived exosomes contributes to tumor growth in the tumor microenvironment of high-grade serous ovarian carcinoma (El-Arabey et al., 2020). Exosomal miRNA confers chemo resistance and overcome tumor progression in ovarian cancer (Kanlikilicer et al., 2018). However, the treatment of chemotherapy-induced POF mice by ADSCs-Exo remains unclear.

The AMPK/mTOR pathway plays a critical role in the regulation of cell autophagy, which may be involved in the development of POF. It has been reported that AMPK/mTOR pathway regulates oocyte aging and female reproduction (Wang et al., 2021a). AMPK is a member of the metabolite-sensing protein kinase family, which can interfere with mTOR activation and then contributes to the metabolic gating of fertility (Bertoldo et al., 2015). Overexpression of mTOR signaling can impair the interaction of cumulus cells, lead to insulin resistance, and affect the growth of follicles directly, leading to POF (Liu et al., 2018). For instance, bisphenol A promotes autophagy and induces AMPK/mTOR signaling pathway in ovarian GCs (Lin et al., 2021). Likewise, Liu et al. (2020b) suggested that nonylphenol promotes both apoptosis and autophagy of GCs cells in rat ovary, which may be related to the activation of reactive oxygen species (ROS)-dependent AMPK/mTOR pathway. In addition, Jin et al. (2019) clarified that ADSCs-Exo can enhance autophagy and reduce podocyte damage by inhibiting the activation of AMPK/mTOR pathway. Meanwhile, ADSCs-Exo improves the chemosensitivity of liver cancer through the AMPK/mTOR pathway (Lou et al., 2020). Further investigations are needed to elucidate whether the action mechanism of ADSCs-Exo involves the AMPK/mTOR pathway in chemotherapeutic drug-induced POF mice.

In view of the above research basis, we suppose that ADSCs-Exo may alleviate chemotherapy-induced POF by inhibiting GC apoptosis and autophagy, which may be involved in the AMPK/mTOR pathway. Therefore, this study aims to explore the therapeutic effects and mechanisms of ADSCs-Exo on chemotherapy-induced POF. Our work lays a solid foundation for the application of ADSCs-Exo in the chemotherapy for POF.

Materials & Methods

Establishment of POF mouse model

Female C57BL/6 mice (n = 36, 18–22 g) aged 6-8 weeks were purchased from the experimental animal center of Inner Mongolia People’s Hospital and acclimated to the conditions of a room at a humidity level of 44 ± 2% humidity and 12 h light/12 h dark cycle at 23 ± 3 °C for one week. Food and water were provided ad libitum. Mice with normal estrous cycle were selected by vaginal exfoliated cell smear. The POF mouse model was established by intraperitoneal injection of 50 mg/kg cyclophosphamide (a chemotherapeutic drug) on the first day, and then continuous injections of 8 mg/kg/day in the next 14 days (Yuan et al., 2022). Vaginal exfoliated cell smear shows the disordered estrous cycle in mice, indicating that the POF mouse model is established successfully (Guo et al., 2019). After the model was established, 15 mg/kg of 3-methyladenine (3-MA, an autophagy inhibitor) dissolved in dimethyl sulfoxide (DMSO) was intraperitoneally injected into POF mice twice a week. The wild-type (WT) group consisted of normal, untreated mice. On the 7th and 15th days after 3-MA treatment, all mice were sacrificed by cervical dislocation. Then, ovaries were removed, and their wet weights were recorded. This study was approved by the Institutional Animal Care and Use Committee of Inner Mongolia People’s Hospital (No. 2020-043).

Separation and identification of ADSCs-Exo

Mouse ADSCs (American Type Cell Culture) were routinely cultured in DMEM/F12 medium. Extraction of ADSCs-Exo was performed as previously described with slight modification (Zhou et al., 2022). Cells of 3–5 generations with good growth were cultured in medium without serum for 24 h. When cell confluence reached 80%, cells were washed with PBS thrice and cultured in fresh medium at 37 °C with 5% CO2 for another 48 h. The cell supernatant was collected after centrifuging for 40 min at 4 °C in an ultrafiltration concentration tube. The exosome-containing concentrate was obtained by centrifuging at 4,000 × g for 40 min and then transferred to a sucrose/heavy water density pad at a concentration of 30%. After sterile membrane filtration and decontamination, ADSCs-Exo were collected from the bottom by centrifuging at 100,000 × g for 120 min at 4 °C. Morphological characteristics of exosomes were observed by transmission electron microscopy (TEM), and the particle-size was analyzed by a Flow Nanoanalyzer (NanoFCM, Nottingham, UK). The protein expression levels of CD63, CD81, TSG101, and HSP70, biomarkers for exosomal surfaces, were assessed through western blot analysis in ADSCs extracted samples, utilizing phosphate-buffered saline (PBS) as the control.

ADSCs-Exo treatment

POF mice received 100 µL of PBS, ADSCs, Exo, Exo + AICAR (0.5g/kg, an AMPK activator), or Exo + Rapa (8 mg/kg, an inducer of mTOR-mediated autophagy) suspension via tail vein injection utilizing a one mL syringe. Mice were injected once every other day from the first day of modeling for 2 weeks. The mice were weighed, and eyeball blood samples were taken on the 7th (n = 3 each group) and 15th (n = 6 each group) day. Mice were sacrificed by cervical dislocation on the 7th and 15th days, and their ovaries were taken out and weighed.

Hematoxylin and eosin (HE) staining

HE staining was conducted by referring to a previously reported method (Wen et al., 2022). Ovarian tissues were collected from mice on the 7th and 15th day after treatment and fixed with 4% paraformaldehyde for 24 h. After gradient alcohol dehydration, transparency, wax dipping, and embedding, tissues were sliced to 5 µm-thick sections. After baking at 60 °C for 2 h, dewaxing with xylene, and hydration with gradient alcohol, sections were stained with hematoxylin aqueous solution for 30 s. Then, sections were stained with eosin for 2 min and then observed under a light microscope (Leica, Wetzlar, Germany).

Enzyme-linked immunosorbent assay (ELISA)

On the 7th and 15th day, eye peripheral blood of mice was taken and centrifuged at 1500 × g at 4 °C for 15 min to obtain the serum supernatant. Levels of serum oxidative stress factors (superoxide dismutase (SOD), malondialdehyde (MDA), and ROS) and serum ovarian function related factors (estradiol (E2) and FSH) in each group were measured by corresponding ELISA kits. Experimental process was operated in strict accordance with the instructions of kits as follows: the SOD kit (SBJ-M0412; Nanjing SenBeiJia, Nanjing, China), the MDA kit (KS13329; Shanghai Keshun Biology, Shanghai, China), the ROS kit (SBJ-M0608; Nanjing SenBeiJia), the E2 kit (E-EL-0150l, Elabscience, Wuhan, China), and the FSH detection kit (E-EL-M0511c; Elabscience).

Immunofluorescence

Ovarian tissues were fixed with 4% paraformaldehyde for 24 h at room temperature. Tissue sections underwent microwave-assisted antigen retrieval at 92–96 °C for 15 min, followed by blocking with 5% BSA at 37 °C for 1 h. The sections were then incubated with FSHR antibody (1:200; AF5242; Affinity, USA) overnight at 4 °C, followed by a 1 h incubation at 37 °C with Goat Anti-Rabbit IgG H&L/AF488 secondary antibody (1:200; bs-0295G-AF488; Bioss, China). Finally, the sections were counterstained with DAPI and visualized under a fluorescence microscope (BX53; Olympus, Tokyo, Japan).

TUNEL assay

Ovarian tissues were collected from mice on the 15th day for TUNEL detection (Luo et al., 2022). Ovarian tissue sections in each group were routinely dewaxed and hydrated, followed by the addition of 50 µL proteinase K working solution for digesting at 37 °C for 30 min. Sections were incubated with 5 µL terminal deoxynucleotidyl transferase (TdT) enzyme, 45 µL fluorescence labeling solution, and 50 µL TUNEL detection solution (Beyotime, Jiangsu, China) at 37 °C for 60 min without light. After rinsed with PBS for three times, 4′,6-diamidino-2-phenylindole (DAPI) dye was dropped on sections and incubated for 10 min at ambient temperature. Sections were sealed with anti-fluorescence quenching sealing solution and then photographed under a fluorescence microscope (Olympus, Japan).

Transmission electron microscopy (TEM)

Ovarian tissues were collected from mice on the 15th day after treatment and subjected to TEM (Dou et al., 2022). Tissues were cut into 1 µm sections and fixed in 2.5% precooled glutaraldehyde solution at 4 °C overnight. After rinsed with PBS thrice, the ovarian tissues were stained with 1% osmic acid, dehydrated with gradient ethanol and 90% acetone, and embedded in ultra-thin sections. After toluidine blue staining, the ovarian tissues were examined under a light microscope (Olympus, Japan). The ultrastructure of ovarian GCs and the changes in mitochondrial autophagy were photographed by TEM (Shanghai Weihan Photoelectric Technology, Shanghai, China).

Cell culture and treatment

Human ovarian granulosa-like tumor (KGN) cell line (Feiya Biotechnology Co., Ltd., Haian, China) were used for in vitro study. KGN cells were cultured in Roswell Park Memorial Institute (RPMI) 1640 Medium (Thermo Fisher Scientific, Waltham, MA, USA) with 10% fetal bovine serum (Gibco) in a humidified incubator at 37 °C with 5% CO2. Cells were divided into four groups: (1) control group (no treatment); (2) cyclophosphamide (CTX) group, in which cells were treated with 250 µM CTX for 48 h to establish POF cell model (Liu et al., 2022); (3) CTX + ADSCs-Exo group, in which cells were co-treated with 250 µM CTX and 10 µg/mL ADSCs-Exo for 48 h; (4) CTX + ADSCs-Exo + Rapa (rapamycin) group, in which cells were pre-treated with 5 µM rapamycin for 1 h and then co-cultured with 250 µM CTX and 10 µg/mL ADSCs-Exo for 48 h.

Cell counting kit (CCK)-8 assay

KGN cell viability was assessed using CCK-8 assay (Yeasen, Shanghai, China). Cells were seeded into a 96-well plate (2 × 103 cells/well). Next day, CCK-8 solution (10 µL) was added to each well for incubation for 48 h. The OD450 nm values were measured using a microplate reader (Bio-Tek, Wuxi City, China).

Flow cytometry

KGN cell apoptosis was detected as previously described (Lin et al., 2021). Cells were cultured in six-well plates (5 × 105 cells/well) overnight. Then, cells were collected and resuspended in binding buffer, followed by incubating with Annexin V-FITC and PI (Beyotime, Jiangsu, China) for 15 min in the dark. Subsequently, cell apoptosis was detected by flow cytometry (BD FACSCalibur).

Western blotting

Western blotting was performed as described elsewhere (Guo, Zhu & Sun, 2020). Total protein was extracted from ovarian tissues and KGN cells using radioimmunoprecipitation (RIPA) lysis solution containing 1% phenylmethyl sulfonyl fluoride (PMSF) and then was quantified with a bicinchoninic acid (BCA) protein concentration kit (Beyotime, China). Proteins were separated by 10% sodium dodecyl sulphate-polyacrylamide gel electrophoresis (SDS-PAGE) and then transferred to polyvinylidene fluoride (PVDF) membrane (Beyotime, Jiangsu, China) via electrophoresis under the condition of 200 mA constant current for 90 min. After blocking with 10% skim milk (Beyotime, Jiangsu, China) for 2 h, membranes were incubated with primary antibodies overnight at 4 °C. Then, membranes were incubated with goat anti-mouse IgG secondary antibody (1:2,000, Abcam, Cambrdige, UK) at room temperature for 2 h. Proteins were developed with ECL chemiluminescence solution and exposed in Alpha InnotechFluorchem SP fluorescence chemiluminescence gel image analysis system (Invitrogen, Waltham, MA, USA). Relative protein expression was calculated by normalizing to GAPDH. The primary antibodies used in this study were the AMPK antibody (1:1,000, ab32047, Abcam, Cambridge, UK), p-AMPK antibody (1:1,000, ab32047, Abcam), mTOR antibody (1:1,000, CSB-PA208208; Cusabio, Hubei, China), p-mTOR antibody (1:1,000, CSB-PA271384; Cusabio), Beclin-1 antibody (1:1,000, ab210498; Abcam), LC3II/LC3I antibody (1:1,000, 12741T; CST), Bcl-2 antibody (1:1,000, ab182858; Abcam), CD63 antibody (1:1,000, ab217345; Abcam), CD81 antibody (1:1,000, DF2306; Affinity, Cincinnati, OH, USA), TSG101 antibody (1:1,000, DF8427; Affinity), HSP70 antibody (1:1,000, AF5466, Affinity) and the GAPDH antibody (1:1,000, ab245355; Abcam).

Statistical analysis

All data were presented in the form of mean ± standard deviation. Animal experiments had six biological and three technical replicates; cell experiments had three of each. Differences among groups were compared by one-way ANOVA, followed by Tukey’s test. All statistical analysis was completed on the GraphPad 7.0 (GraphPad Software, La Jolla, CA, USA). P < 0.05 was considered to indicate a statistically significant difference between groups.

Results

Exosomes derived from stem cells have been reported to be effective in treating POF, since they act as an essential messenger for intracellular communication (Qu et al., 2022). In this study, we explored the pathological role of autophagy in POF. Meanwhile, the therapeutic effects and mechanisms of ADSCs-Exo were investigated to be related to the regulation of autophagy and the AMPK/mTOR pathway.

Inhibition of autophagy promoted ovarian growth and functional recovery, and suppressed oxidative stress in POF mice

In order to evaluate the effect of autophagy on ovarian function in POF, model mice were induced by cyclophosphamide and treated with autophagy inhibitor 3-MA (Fig. 1). Compared to WT mice, ovarian tissues of POF mice were smaller and had lower weight. Treatment of 3-MA for 15 days recovered the ovarian growth of POF mice (p < 0.01, Fig. 2A). Subsequently, the effects of autophagy inhibition on pathological tissue injury in POF mice were evaluated. HE staining showed a decrease of sinus follicles, GC layer, and corpus luteum, and an increase of atretic follicles in the POF group compared to the WT group. These pathologies were alleviated by 3-MA treatment in POF mice. In addition, the pathological injury was more obvious on the 15th days than on the 7th days after 3-MA treatment (Fig. 2B). On the other hand, ELISA was used to evaluate the levels of ovarian function related factors E2 and FSH. In POF, follicles, estrogen, and progesterone dramatically decrease in ovaries, resulting in an increased serum FSH level and a decreased E2 level (Fu et al., 2017). Our results showed that the levels of E2 and FSH in the POF group were significantly decreased and increased compared with that in the WT group, respectively (p < 0.01). 3-MA intervention for 15 days markedly enhanced E2 level and down-regulated FSH level in POF mice (p < 0.05, Fig. 2C). Furthermore, MDA, ROS, and SOD are critical biomarkers of oxidative stress. ELISA showed that on the 15th day, the contents of SOD in the POF group were substantially decreased, whereas ROS and MDA levels were prominently increased compared with that in the WT group (p < 0.01). 3-MA intervention enhanced SOD, and declined ROS and MDA levels in POF mice on the 15th day (p < 0.05, Fig. 2C).

Figure 1 Schematic presentation of the premature ovarian failure (POF) animal model construction and 3-MA treatment.

Figure 2 Inhibition of autophagy reduced oxidative stress and recovered ovarian function in premature ovarian failure (POF) mice.

(A) Ovarian tissues were collected on the 7th and 15th days and weighed. Scale bar: 5 mm. (B) Histological changes of ovary were observed by hematoxylin-eosin (HE) staining. Scale bar: 20 µm. (C) ELISA was used to evaluate the levels of serum ovarian function related factors (E2 and FSH) and oxidative stress factors (MDA, ROS, and SOD). **p < 0.01 compared with the WT group, #p < 0.05 and ##p < 0.01 compared with the POF group.

Inhibition of autophagy alleviated GC apoptosis in POF mice

Apoptosis of ovarian GCs is the initiating factor of POF (Liu et al., 2021). We found that the expression level of FSHR, a known GC biomarker, was significantly reduced in the POF group compared to the WT group (p < 0.01). Treatment with 3-MA successfully upregulated the FSHR level in the POF group, as evidenced by the results (p < 0.01, Fig. S1). TUNEL showed that the apoptosis rate of GCs in the POF group was higher than that in the WT group, which was decreased by 3-MA treatment (Fig. 3A). In addition, TEM was used to observe the ultrastructure of ovarian GCs and the changes of mitochondrial autophagy. Results showed that compared with the WT group, the nuclei of oocytes changed from prismatic to vacuolar in the POF group. Also, the number of mitochondria in GCs decreased, while the number of autophagosomes increased in the POF group. Administration of 3-MA recovered the morphology of GCs and reduced the number of autophagosomes (Fig. 3B). Bcl-2 and Beclin-1 are key regulators of autophagy (Hill, Wrobel & Rubinsztein, 2019; Xu et al., 2020), and LC3II/LC3I can target autophagosomes to mitochondria and induce mitochondrial autophagy (Unal et al., 2021). Western blotting showed that, compared with the WT group, Beclin-1 and LC3II/LC3I levels were significantly increased in the POF group, while Bcl-2 was decreased (p < 0.01). 3-MA treatment markedly downregulated the protein levels of Beclin-1 and LC3II/LC3I and upregulated Bcl-2 level in POF mice (p < 0.05, Fig. 3C).

Figure 3 Inhibition of autophagy alleviated pathological injury and granulosa cell (GC) apoptosis in POF mice by retarding the AMPK/mTOR pathway.

(A) Ovarian tissue samples were collected on the 15th day, and the apoptosis of ovarian GCs was detected by TUNEL staining. Scale bar: 20 µm. (B) The ultrastructure and intracellular autophagy of cells in ovarian tissue were observed by transmission electron microscope. Scale bar: 2 µm. (C) Levels of autophagy-related genes (Beclin-1 and LC3II/LC3I), apoptosis-related protein Bcl-2, and pathway-related proteins (p-AMPK/AMPK and p-mTOR/mTOR) were detected by western blotting. **p < 0.01 compared with the WT group. #p < 0.05 and ##p < 0.01 compared with the POF group. Ovarian tissue samples were collected on the 15th day.

Inhibition of autophagy reduced GC apoptosis by retarding the AMPK/mTOR pathway in POF mice

Studies have confirmed that mTOR can act as an “activity switch” in the regulation of autophagy (Kazibwe et al., 2020; Wang & Zhang, 2019). The AMPK/mTOR pathway plays an important regulatory role in the process of autophagy (Li et al., 2019). In addition, the AMPK/mTOR pathway implicates oocyte senescence and female reproduction (Wang et al., 2021a). We further explored the role of AMPK/mTOR pathway during 3-MA treatment for POF mice. Western blotting showed that p-AMPK/AMPK expression was increased, and p-mTOR/mTOR level was significantly decreased in POF mice compared with that in WT mice, while reversed by 3-MA treatment (p < 0.05, Fig. 3C).

ADSCs-Exo ameliorated chemotherapy-induced pathological damage in POF mice

Subsequently, we evaluated the effect of ADSCs-Exo in the treatment of POF. In the present study, exosomes isolated from ADSCs exhibited a circular morphology under TEM, with the average particle-size of 56.67 nm (Fig. 4A). Western blotting showed that the surface specific marker proteins CD63, CD81, TSG101, and HSP70 were expressed in ADSCs-Exo compared with that in the control group (Fig. 4B). Firstly, the morphology of ovarian tissue was observed after ADSCs-Exo treatment. The results showed that on the 15th day, ovarian size and weight in the ADSCs group and Exo group were substantially increased compared with the POF group (p < 0.05, Fig. 4C). Furthermore, HE staining showed that the number of follicles in the ADSCs group and Exo group was increased, while atresia follicles were decreased (Fig. 4D).

Figure 4 Adipose mesenchymal stem cell-derived exosomes (ADSCs-Exo) ameliorated chemotherapy-induced pathological damage in POF mice.

(A) Transmission electron microscope scanning showed the morphology of ADSCs-Exo, and the particle-size was analyzed by a Flow Nanoanalyzer. Scale bar: 0.5 µm. (B) Surface specific marker of exosomes (CD63, CD81, TSG101, and HSP70) was detected by western blotting. (C) Ovarian tissues were collected on the 7th and 15th days and weighed. Scale bar: 5 mm. (D) Histological changes of ovary were observed by HE staining. Scale bar: 20 µm. **p < 0.01 compared with the WT group, ##p < 0.01 compared with the POF group, and ˆ ˆp < 0.01 compared with the ADSCs group.

ADSCs-Exo inhibited GC apoptosis and autophagy by impeding the AMPK/mTOR pathway

Then, ELISA showed that on the 15th day, compared with the POF group, ADSCs and Exo markedly increased E2 level and decreased the contents of FSH (p < 0.05, Fig. 5A). Likewise, ADSCs and Exo markedly increased SOD level and decreased the contents of ROS and MDA in POF mice (p < 0.05, Fig. 5A). On the other hand, TUNEL showed that, ADSCs-Exo treatment reduced the apoptosis rate of GCs in POF mice (Fig. 5B). Subsequently, western blotting presented that, compared with the POF group, the protein levels of Beclin-1 and LC3II/LC3I in ADSCs and Exo group were dramatically downregulated, and Bcl-2 level was upregulated. Finally, the role of the AMPK/mTOR pathway in ADSCs-Exo treatment for POF mice was verified. Western blotting showed that p-AMPK/AMPK level was decreased, and p-mTOR/mTOR was significantly increased in ADSCs and Exo groups compared with that in the POF group (p < 0.05, Figs. 5C and 5D). Administering AICAR, an AMPK activator, reversed the inhibitory effect that ADSCs-Exo had on POF (p < 0.05, Fig. 6). Furthermore, administering Rapa—an inducer of mTOR-mediated autophagy—reversed the protective effect exerted by ADSCs-Exo against POF (p < 0.05, Figs. 7A and 7B).

Figure 5 ADSCs-Exo inhibited GC apoptosis and autophagy by suppressing the AMPK/mTOR pathway.

(A) ELISA was used to evaluate the levels of E2, FSH, MDA, ROS, and SOD. (B) Ovarian tissue samples were collected on the 15th day, and the apoptosis of ovarian GCs was detected by TUNEL staining. Scale bar: 20 µm. (C–D) The protein levels of Beclin-1, LC3II/LC3I, Bcl-2, p-AMPK, and p-mTOR were detected by western blotting. **p < 0.01 compared with the WT group, #p < 0.05 and ##p < 0.01 compared with the POF group, and ˆ ˆp < 0.05 and ˆ ˆp < 0.01 compared with the ADSCs group.

Figure 6 ADSCs-Exo inhibited GC autophagy by suppressing the AMPK/mTOR pathway.

The protein levels of p-AMPK/AMPK, p-mTOR/mTOR, Bcl-2, Beclin-1, and LC3II/LC3I were detected by western blotting. **p < 0.01 compared with the PBS group, #p < 0.05 and ##p < 0.01 compared with the Exo group.

Figure 7 ADSCs-Exo alleviated pathological injury in POF mice by inhibiting mTOR-mediated autophagy.

(A) Ovarian tissues were collected on the 7th and 15th days and weighed. Scale bar: 5 mm. (B) Histological changes of ovary on the 15th day were observed by hematoxylin-eosin (HE) staining. Scale bar: 20 µm.

Further, CTX-treated KGN cells were treated with ADSCs-Exo or/and Rapa to confirm the mechanism of ADSCs-Exo inhibiting GC autophagy in POF. CTX (an alkylating agent) has a pro-apoptotic adverse effect on ovarian GCs (Li et al., 2021b), which was used to treat human ovarian granulosa-like tumor (KGN) cells to construct the cell model of POF. Using the CCK-8 assay, we determined that the IC50 value for ADSCs-Exo in treating CTX-treated KGN cells was 10 µg/Ml (Fig. 8A). Consequently, a concentration of 10 µg/mL for ADSCs-Exo was utilized in subsequent experiments involving the treatment of CTX-treated KGN cells. CCK-8 and flow cytometry showed that CTX inhibited the viability and promotes the apoptosis of KGN cells (p < 0.01). ADSCs-Exo treatment alleviated CTX-induced apoptosis, whereas the addition of rapamycin repressed the effect of ADSCs-Exo on CTX-treated KGN cells (p < 0.05, Figs. 8B and 8C). Regarding the mechanism, CTX upregulated the levels of Beclin-1 and LC3II/LC3I, and downregulated the Bcl-2 level in KGN cells (p < 0.01). ADSCs-Exo administration reduced CTX-induced autophagy, evidenced by the decreased Beclin-1 and LC3II/LC3I as well as the increased Bcl-2 level (p < 0.01). The addition of rapamycin weakened the inhibitory effect of ADSCs-Exo on autophagy in CTX-treated KGN cells (p < 0.05, Fig. 8D). In addition, expression of p-AMPK was upregulated, and that of p-mTOR was downregulated in CTX-treated KGN cells (p < 0.01). ADSCs-Exo treatment decreased the p-AMPK/AMPK level and increased the p-mTOR/mTOR level in CTX-treated KGN cells, whereas rapamycin addition reversed the effect of ADSCs-Exo (p < 0.05, Fig. 8D).

Figure 8 ADSCs-Exo inhibited AMPK/mTOR-mediated autophagy in cyclophosphamide (CTX)-treated human ovarian granulosa-like tumor (KGN) cells.

(A) IC50 value for ADSCs-Exo treating CTX-treated KGN cells was identified by cell counting kit (CCK)-8 assay. (B) Cell viability was detected by CCK-8 assay. (C) Cell apoptosis was assessed by flow cytometry. (D) The protein levels of Beclin-1, LC3II/LC3I, Bcl-2, p-AMPK/AMPK, and p-mTOR/mTOR were detected by western blotting. KGN cells were treated with 250 µM CTX, 10 µg/mL ADSCs-Exo, or/and 5 µM rapamycin (Rapa). **p < 0.01 compared with the control group, ##p < 0.01 compared with the CTX group, and ˆp < 0.05 and ˆ ˆp < 0.01 compared with the CTX + ADSCs-Exo group.

Discussion

POF refers to the weakening or even failure of ovarian function in women before the age of 40. In recent years, the incidence rate of POF has gradually increased and showed a younger trend, seriously affecting the patient’s health and quality of life (Lin et al., 2017). Chemotherapeutic drugs can cause serious damage to ovarian physiology, inducing conditions such as follicular atresia, GC apoptosis, abnormal hormone secretion, oocyte apoptosis, and fibrosis, which are notable features of POF (Lande et al., 2017; Winship et al., 2018). Autophagy can cause type II programmed cell death, therefore, autophagic death of granulosa cells may be a potential factor contributing to POF (Yin et al., 2020). Earlier studies have shown that BMSC-derived exosomes are protective against POF in rats (Liu et al., 2020a; Sun et al., 2019). In the current study, cyclophosphamide (a chemotherapeutic drug) was used to induce POF in mice to explore the therapeutic efficacy and mechanisms of ADSCs-Exo in POF. We found that ADSCs-Exo inhibited apoptosis and autophagy of GCs in POF by affecting the AMPK/mTOR pathway.

Under various cellular stress conditions, autophagy prevents cell damage and promotes survival through regulating highly conserved catabolic pathways. Autophagy also plays an important role in POF (Li et al., 2017). In this study, to evaluate the effects of autophagy in POF, model mice of POF were established by cyclophosphamide induction and then treated with an autophagy inhibitor 3-MA. We found that 3-MA treatment decreased the apoptosis rate of GCs, downregulated the protein levels of Beclin-1 and LC3II/LC3I, and upregulated the Bcl-2 levels in POF mice. These results suggest that inhibition of autophagy can alleviate GC apoptosis and autophagy induced by POF. Oxidative stress also mediates the progression of POF and GCs autophagy (Li et al., 2021a; Liu et al., 2021). Wang et al. (2021a) suggested that oxidative stress and possible oocyte aging contribute to the development of POF. Yan et al. (2018) demonstrated that curcumin can alleviate POF by inhibiting oxidative stress. ROS, MDA, and SOD can reflect the level of oxidative stress (Bilgen et al., 2019; Wang et al., 2020). We found that the contents of MDA and ROS were markedly increased, and SOD level was decreased in POF mice. However, 3-MA intervention increased SOD levels and decreased the MDA and ROS levels in POF mice, suggesting that inhibition of autophagy is able to suppress oxidative stress in POF. Furthermore, POF is accompanied by an increase of serum FSH level and a decrease of E2 level (Wang et al., 2019a; Yan et al., 2018). In the current study, 3-MA treatment downregulated FSH level and upregulated E2 level in POF mice. This implies that inhibition of autophagy is able to restore ovarian function in POF. HE staining showed that 3-MA treatment improved the ovarian morphology and significantly increased the ovarian weight of POF mice. This suggests that inhibition of autophagy can alleviate pathological tissue damage in POF.

Furthermore, the potential mechanism of autophagy in POF was explored. Wang et al. (2021a) confirmed that the AMPK/mTOR pathway is related to oocyte aging during POF. At present, western blotting showed that the levels of p-AMPK were significantly increased and the levels of p-mTOR were decreased in POF mice; these changes were reversed by 3-MA treatment. This indicates that inhibition of autophagy reduced GC apoptosis and autophagy by retarding the AMPK/mTOR pathway.

Previous studies have confirmed that ADSCs can effectively alleviate GC apoptosis and autophagy (Wang et al., 2021b). We found that ADSCs-Exo intervention increased SOD levels and decreased MDA and ROS levels in POF mice. This result suggests that ADSCs-Exo inhibits oxidative stress in POF mice. In addition, ADSCs-Exo treatment downregulated FSH level and upregulated E2 level in POF mice, suggesting that ADSCs-Exo improves ovarian function and morphology. On the other hand, HE staining showed that ADSCs-Exo alleviated the pathological damage of ovarian tissue in POF mice. In addition, TUNEL results showed that ADSCs-Exo treatment reduced the apoptotic rate of GCs in POF mice. Moreover, ADSCs-Exo downregulated Beclin-1 and LC3II/LC3I protein levels and increased Bcl-2 expression in POF mice. These results suggest that ADSCs-Exo can inhibit autophagy of GCs in POF. Furthermore, we evaluated the role of the AMPK/mTOR pathway in ADSCs-Exo treating POF. Western blotting showed that ADSCs-Exo intervention downregulated the p-AMPK level and upregulated the p-mTOR level in POF mice, indicating that ADSCs-Exo inhibit the AMPK/mTOR pathway in POF. Furthermore, the mechanism of ADSCs-Exo against POF involving autophagy and AMPK/mTOR pathway was validated in vitro. KGN cells were treated with CTX to establish cell model of POF. Similar to in vivo results, ADSCs-Exo administration enhanced the viability and reduced the apoptosis of CTX-treated KGN cells. However, rapamycin (an inducer of mTOR-mediated autophagy) addition weakened the protective effect of ADSCs-Exo on CTX-treated KGN cells. Also, ADSCs-Exo inhibited the autophagy and AMPK/mTOR pathway in CTX-treated KGN cells, evidenced by the increased Bcl-2 and p-mTOR levels, as well as the decreased Beclin-1, LC3II/LC3I, and p-AMPK levels. Rapamycin addition reversed the inhibitory effect of ADSCs-Exo on AMPK/mTOR-mediated autophagy. These results further confirmed that ADSCs-Exo inhibits POF via blockage of autophagy and AMPK/mTOR pathway.

In our study, we have demonstrated that ADSCs-derived exosomes not only modulate autophagy through the AMPK/mTOR pathway but also facilitate a series of reparative processes including reducing apoptosis and fostering the regeneration of ovarian tissues. These exosomes are naturally derived, potentially offering a better safety profile compared to synthetic inhibitors like 3-MA. Furthermore, exosomes can encapsulate a wide variety of bioactive molecules, creating a multifaceted approach to therapy that may address a broader range of pathological changes in POF, going beyond what 3-MA can offer.

There are also deficiencies in our study. Firstly, to avoid the immune rejection of allograft, adipose-derived stem cells were isolated from mice to treat POF in mice. The role of exosomes is needed to be further confirmed using human ADSCs. Secondly, this study explored the role and mechanism of ADSCs-Exo in chemotherapeutic drug-induced POF in vivo, which needs to be further confirmed in clinic. Moreover, we recognize the complexity of exosomal contents and the potential for different regulatory mechanisms in the animal and cell models used in our research. Future studies will delve deeper into exploring these aspects to provide a more comprehensive understanding. Furthermore, Wang et al. (2019b) found that the chemical pesticide diazinon induced POF and follicular atresia, and promoted GC apoptosis and autophagy by inhibiting PI3K-AKT signaling. It is needed to be further confirmed whether there are other mechanisms of ADSCs-Exo treating chemotherapeutic drug-induced POF.

Conclusion

In summary, our study demonstrated that ADSCs-Exo exerts the therapeutic effect on chemotherapeutic drug-induced POF by suppressing the apoptosis and autophagy of GCs. Further, the potential mechanism of ADSCs-Exo against POF was found to be involved in the inhibition of the AMPK/mTOR pathway. These findings lay a solid foundation for the application of ADSCs during chemotherapy of POF.

Supplemental Information

Data S1 Raw data

Click here for additional data file.

Supplemental Information 2 Threefold repetition of western blots

Click here for additional data file.

Supplemental Information 3 Original images of western blot

Click here for additional data file.

Figure S1 Inhibition of autophagy increased the expression of granulosa cell (GC) biomarker FSHR in ovarian tissues of POF mice

The expression of FSHR was measured by immunofluorescence. Scale bar: 20 µm. ** p < 0.01 compared with WT group, ## p < 0.01 compared with POF + DMSO group.

Click here for additional data file.

Supplemental Information 5 ARRIVE checklist

Click here for additional data file.

Additional Information and Declarations

Competing Interests

Author Contributions

Animal Ethics

Data Availability

The authors declare there are no competing interests.

Yu Ren conceived and designed the experiments, performed the experiments, analyzed the data, prepared figures and/or tables, authored or reviewed drafts of the article, and approved the final draft.

Jinying He conceived and designed the experiments, performed the experiments, analyzed the data, prepared figures and/or tables, and approved the final draft.

Xiao Wang performed the experiments, analyzed the data, prepared figures and/or tables, authored or reviewed drafts of the article, and approved the final draft.

Hongyu Liang performed the experiments, analyzed the data, authored or reviewed drafts of the article, and approved the final draft.

Yuzhen Ma conceived and designed the experiments, authored or reviewed drafts of the article, and approved the final draft.

The following information was supplied relating to ethical approvals (i.e., approving body and any reference numbers):

This study was approved by the Institutional Animal Care and Use Committee of Inner Mongolia People’s Hospital (No. 2020-043).

The following information was supplied regarding data availability:

The raw data is available in the Supplemental File.

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
