# Peer review of "Exosomes from adipose-derived stem cells alleviate premature ovarian failure via blockage of autophagy and AMPK/mTOR pathway"

_PeerJ, doi:10.7717/peerj.16517_

## Round 0.1 · original submission · Major Revisions

Please carefully read the comments and suggestions from the reviewers and provide your point-by-point response.

·

Basic reporting

1. The image resolution is too low, please provide a higher resolution image.
2. Fig 1A, the 7-day POF group looks a bit larger than the WT group tissue, why is the quantification result opposite?
3.Fig1A, p-value, Please indicate in the figure which two groups are the result of comparison.
4.Fig1, according to the authors' text, the DMSO group should be POF+DMSO, and the 3-MA group should be POF+3-MA.
5.Fig1, it is suggested to draw an additional graph to indicate the time and frequency of various drugs in different groups.

Experimental design

6. Fig2A, there are many cells in the ovary, how did you locate the GCs? Is it necessary to stain one more marker protein of GCs.
7. Fig2 and fig3 suggest to combine into one graph
8. fig3, please label A and B in the graph
9.Fig3, please add the protein expression of AMPK and mTOR
10.Fig4B, please indicate what proteins were extracted from the control group
11. Fig4C, please indicate what interventions were done in the PBS group, ADSCs group and Exo group respectively, and indicate the specific time and frequency of the interventions.
12. Fig5, these experimental data can only show that ADSCs-Exo can inhibited GC apoptosis and autophagy, and also suppressing the AMPK/mTOR Pathway, but can not prove that “ADSCs-Exo inhibited GC apoptosis and autophagy by suppressing the AMPK/mTOR pathway.” You need to do a rescue experiment to prove your claim.

Validity of the findings

13. fig6A, please indicate what the grouping represents.
14. fig6B, please show the ratio of the four quadrants in larger font.
15. fig6C, please add the expression of AMPK and mTOR.
16. The authors demonstrated that both Exo and 3-MA can affect premature ovarian failure, but did not prove at the animal level that Exo affects premature failure by autophagy, and more rescue experiments are needed to prove this conclusion.
17. Please add scale bar in all the pictures
18. Please add a ruler next to the pictures that involve real objects
19. Discussion:The premature ovarian failure mentioned in the previous section is caused by chemotherapy, but chemotherapy is not mentioned here, so please be consistent.

Reviewer 2 ·

Basic reporting

no comment

Experimental design

no comment

Validity of the findings

no comment

Additional comments

Premature Ovarian Failure (POF) is a common clinical disease with complex molecular mechanisms, hormone replacement is a widely used therapeutic strategy; however, it does not fully recover ovarian function and fertility.
This study was designed and conducted at the mouse model and cellular level to investigate the effects of adipose-derived stem cell-derived exosome (ADSCs-Exo) against POF, the results indicated that ADSCs-Exo alleviated pathological injury, improved ovarian morphology and function, and reduced oxidative stress in POF mice, and ADSCs-Exo inhibited GC apoptosis and autophagy, this study also suggested that ADSCs-Exo against POF by inhibiting AMPK/mTOR-mediated autophagy.
While the studies are thorough and provide significant insight into the interaction between apoptosis and autophagy, there are some areas that require revisions.

1. English language revision:
For example,
Line 94: "ADSCS-Exo" should be expressed as "ADSCs-Exo."
Line 105: "Food and water were provided via free choice."
Line 164: "2.7 ransmission electron microscopy (TEM)" should be "2.7 Transmission electron microscopy (TEM)."
Line 217-221: Is this section necessary?

2. Clarify sentences:
For example,
line 93-95, "The action mechanism of ADSCS-Exo on chemotherapeutic drug-induced POF mice involving in 95 AMPK/mTOR pathway remains to be further elucidated" should be "whether ...?"
line 229 “sinusoidal follicles”, is this refer to "sinus follicles"?
line 231-232 “In addition, the pathological phenomenon was more obvious at 15th days than that at 7th days after 3-MA treatment” is it the right?

Line 93-95: Revise "The action mechanism of ADSCS-Exo on chemotherapeutic drug-induced POF mice involving in 95 AMPK/mTOR pathway remains to be further elucidated" to "Further investigations are needed to elucidate whether the action mechanism of ADSCs-Exo involves the AMPK/mTOR pathway in chemotherapeutic drug-induced POF mice."
Line 229: "sinusoidal follicles" should be revised to "sinus follicles."
Line 231-232: "In addition, the pathological phenomenon was more obvious at 15th days than that at 7th days after 3-MA treatment" should be clarified for accurate understanding.

3. Improve figure clarity:
For example,
Figure 1B: Increase the scale of the figure to improve legibility.
Figure 3: Add appropriate labels (e.g., AB mark) to improve clarity.
Ensure all figures are presented neatly.

4. Others
1) The establishment of POF model is of great significance to this research, please provide more detailed data on ovarian observation after treatment.
2) Describe the process used to determine the optimal treatment concentration of 10 µg/mL ADSCs-Exo, including any pre-testing conducted.
3) The exosomal contents are complex, and since animal models and cell models belong to different species in this study, whether there are different regulatory mechanisms between them needs to be distinguished.

Reviewer 3 ·

Basic reporting

The authors created A POF mouse model via intraperitoneal injection of cyclophosphamide, as well as A POF cell model using cyclophosphamide treated human ovarian granulosa-like tumor (KGN) cells. Then they used adipose mesenchymal stem cells derived exosomes, 3-methyladenine (3-MA) or rapamycin to treat POF mice and cells followed by investigating the autophagy related biological incidents. The results identified that exosomes inhibit POF by blocking autophagy and the AMPK/mTOR pathway.
The whole manuscript was presented logically and clearly with professional English. The introduction and the discussion were sufficient. The results answer the hypotheses.

Experimental design

This study was designed well. The article meets PeerJ standards.

Validity of the findings

The article meets PeerJ standards.

Additional comments

1. I would suggest to remove one of the two “derived” from the title, or changed the title in to” Exosomes derived from adipose mesenchymal stem cells alleviate premature ovarian failure via blockage of autophagy and AMPK/mTOR pathway” as well as the abbreviation of adipose mesenchymal stem cells into AMSC.
2. I would suggest to remove “by inhibiting AMPK/mTOR-mediated autophagy” from line 24 to avoid the repeated statement in the abstract.
3. Whether the nouns of “Beclin” and “Becline” in figure3,5 and6, need to be unified?
4. The statement “Adipose-derived stem cells (ADSCs) are mesenchymal stem cells from adipose tissue” in line 64 is tedious, please edit it.

·

Basic reporting

The manuscript by Yu Ren et. al. evaluates the effect of adipose-derived stem cells secreted exosomes in treating premature ovarian failure (POF). The authors have performed several experiments trying to show that autophagy inhibition either via 3-MA blocking or the use of adipose-derived stem cells exosomes led to improved outcomes of premature ovarian failure by the AMPK/mTOR pathway. It has all the necessary experiments performed, however, there are some concerns that need to be addressed before we can consider it for publication.

1. If POFs can be alleviated via the use of autophagy inhibitors e.g. 3-MA, what is the advantage of using adipose-derived stem cells exosomes in the first place? The authors need to convince the readers that adipose-derived stem cell exosomes provide more benefits over 3-MA treatment.

2. The manuscript has some grammatical/typo errors which need to be addressed.

Overall, it’s a nicely executed work but needs some more insights about its significance and I would recommend it for publication with these minor revisions.

Experimental design

The experimental design was good.

Validity of the findings

no comment

---

## Round 0.2 · Minor Revisions

Please carefully read the comments from the Section Editor and provide your point-by-point responses.

"There are no indications of technical and biological replicate numbers throughout. This is essential.

The exosome characterisation is very limited - a single blot of CD36 is insufficient. Please go to https://www.isev.org/ and adopt their minimum requirements. I suspect the authors are close to this, but it needs to be explicitly stated.

Only a single representative immunoblot is uploaded as supplemental supporting data. All biological replicates should be shown."

Reviewer 2 ·

Basic reporting

no comment

Experimental design

no comment

Validity of the findings

no comment

Additional comments

Basically, the revised version of manuscript was improved well and carefully addressed to raised points , I would agree the manuscript fits for publication in PeerJ.

Reviewer 3 ·

Basic reporting

The authors have revised this manuscript well according to my comments. The revised version seems to be alright for the publication in PeerJ.

Experimental design

no comment

Validity of the findings

no comment

Additional comments

no comment

---

## Round 0.3 · accepted · Accept

The authors have addressed the questions from the section editor and the current version of the manuscript may be considered for publication.